# Strong geometry dependence of the Casimir force between interpenetrated rectangular gratings

Mingkang Wang [1,2,3], L. Tang[1,2,3], C. Y. Ng[1], Riccardo Messina [4,5], Brahim Guizal [5], J. A. Crosse[6,7], Mauro Antezza [5,8], C. T. Chan[1] & H. B. Chan [1,2,3 ✉]

Quantum fluctuations give rise to Casimir forces between two parallel conducting plates, the magnitude of which increases monotonically as the separation decreases. By introducing nanoscale gratings to the surfaces, recent advances have opened opportunities for controlling the Casimir force in complex geometries. Here, we measure the Casimir force between two rectangular silicon gratings. Using an on-chip detection platform, we achieve accurate alignment between the two gratings so that they interpenetrate as the separation is reduced. Just before interpenetration occurs, the measured Casimir force is found to have a geometry dependence that is much stronger than previous experiments, with deviations from the proximity force approximation reaching a factor of ~500. After the gratings interpenetrate each other, the Casimir force becomes non-zero and independent of displacement. This work shows that the presence of gratings can strongly modify the Casimir force to control the interaction between nanomechanical components.

[1] Department of Physics, The Hong Kong University of Science and Technology, Clear Water Bay, Kowloon, Hong Kong, China. [2] William Mong Institute of Nano Science and Technology, The Hong Kong University of Science and Technology, Clear Water Bay, Kowloon, Hong Kong, China. [3] Center for Metamaterial Research, The Hong Kong University of Science and Technology, Clear Water Bay, Kowloon, Hong Kong, China. [4] Laboratoire Charles Fabry, UMR 8501, Institut d'Optique, CNRS, Université Paris-Saclay, 2 Avenue Augustin Fresnel, 91127 Palaiseau Cedex, France. [5] Laboratoire Charles Coulomb (L2C), UMR 5221 CNRS-Université de Montpellier, F-34095 Montpellier, France. [6] New York University Shanghai, 1555 Century Ave, Pudong, 200122 Shanghai, China. [7] NYU-ECNU Institute of Physics at NYU Shanghai, 3663 Zhongshan Road North, 200062 Shanghai, China. [8] Institut Universitaire de France, 1 rue Descartes, F-75231 Paris, France. ✉email: hochan@ust.hk

The prediction of the attractive force between two planar perfect mirrors by Casimir is based on the effect of boundary conditions imposed on the zero-point fluctuations of the electromagnetic field[1]. As the separation between the two flat surfaces is decreased, the Casimir force increases rapidly and monotonically. Lifshitz extended the analysis to real materials by considering the polarization fluctuations within the interacting bodies and calculated the force in terms of the dielectric properties of the material[2,3]. In the past two decades, advances in mechanical transducers and atomic force microscopes have enabled precision measurements of the Casimir force[4–17]. A number of these experiments address important issues such as the role of relaxation at low frequencies in the calculation of the Casimir force[18–20]. Apart from fundamental interest, studies of the Casimir force are also relevant to the fabrication and operation of nanomechanical systems in which the movable components are in close proximity[21–24].

One remarkable property of the Casimir force is its nontrivial dependence on the geometry of the interacting objects. For slight deviations from the parallel-plate configuration, the proximity force approximation (PFA)[25] is often used to estimate the Casimir force. Under the PFA, the surfaces of the two bodies are divided into small parallel plates. The total force is obtained by summing up the local forces between pairs of plates that are assumed to be given by Lifshitz's formula. While the PFA provides a convenient way to estimate the Casimir force for simple geometries, it is not applicable for objects with complicated shapes[26]. The dependence of the Casimir force on geometry and the interplay with optical properties of the material[27–33] opens new opportunities for applications in which the Casimir force needs to be controlled.

The vast majority of experiments on the Casimir force require replacing at least one of the planar surfaces by a sphere to avoid the difficulty of maintaining parallelism between the surfaces at small separations[34]. Other configurations including plate-plate[35] and sphere-sphere[36] have also been measured experimentally. Provided that the radius of the sphere is much larger than the separation, the Casimir force in the sphere-plate geometry can be estimated using the PFA. To reveal the geometry dependence of the Casimir force, it is necessary to introduce nanoscale gratings onto the interacting bodies. Deviations from the PFA were observed in a configuration where the flat surface in the sphere-plate geometry is replaced by silicon or gold gratings[37–39]. The largest deviation observed so far are ~80%[39]. Even though the PFA cannot predict the Casimir force accurately in these experiments, it is computationally undemanding and is useful for a quick estimate of the order of magnitude of the force.

Recent progress in theoretical and numerical methods has enabled the calculation of Casimir forces for objects of arbitrary shapes[34]. A number of groups have developed schemes based on the scattering theory to calculate the Casimir force for gratings[27–33,40]. The accuracy of these calculations improves as the number of Fourier components in the computation is increased. One-dimensional and two-dimensional gratings of different shapes have been extensively considered[41]. If gratings are present on both interacting surfaces, the Casimir force can be calculated provided that the two objects are separated by a planar boundary. In other words, schemes based on the scattering theory are valid as long as the two gratings do not interpenetrate. Apart from the force, the calculations can also yield the Casimir torque between two gratings[42,43]. Such torque between gratings has been predicted to be significantly stronger than those in anisotropic materials that were recently demonstrated in experiments[44].

When nanoscale gratings are present on both surfaces, measurement of the Casimir force poses additional challenges. Other than the usual alignment requirements for flat plates[35,45], the relative orientation and lateral shift between the two gratings also need to be accurately controlled. So far, only one team has measured the Casimir force between two gratings[26]. By imprinting the sinusoidal grating pattern onto a gold sphere and measuring the force in-situ, the lateral Casimir force between the two corrugated surfaces has been demonstrated to deviate significantly from the PFA[46]. The measurement was performed when the two gratings were well-separated from each other without any interpenetration. A prior experiment measured the nonmonotonic Casimir force when two T-shaped protrusions interpenetrate[14]. However, due to the limited resolution of optical lithography in the fabrication process, the protrusions are rounded at the corners. Moreover, there are nonuniformities among the different units, introducing uncertainties so that deviations from the PFA cannot be unambiguously identified. To our knowledge, the strong geometry dependence of the Casimir force in the regime of interpenetration for rectangular gratings remains unexplored.

In this paper, we measure the Casimir force between two rectangular silicon gratings. With the gratings defined in a single electron-beam lithography step, they are accurately aligned so that they interpenetrate as the distance between them is reduced using an on-chip comb actuator[45]. The Casimir force gradient is inferred from the shift in the resonance frequency of a doubly clamped beam that supports one of the gratings. Right before interpenetration occurs, the measured Casimir force is shown to be ~500 times larger than the PFA, yielding a geometry dependence that is about two orders of magnitude stronger than previous experiments[14,37–39,46]. After interpenetration, a novel distance dependence of the Casimir force emerges. The force is shown to be non-zero but independent of displacement. For this geometry, the PFA and the pairwise-additive approximation (PAA) yield different estimates of the Casimir force. Specifically, the PFA and the PAA works well only for the region after and before interpenetration, respectively. The experiment involves a number of improvements to the detection platform to enable the fabrication structures in which, for a certain range of parameters, the PFA breaks down completely and fails to estimate even the order of magnitude of the Casimir force. There is good agreement between measurement and exact calculations using boundary element methods over the entire distance range, including the region where the PFA breaks down.

## Results

**The Casimir force between perfect rectangular gratings**. We consider the Casimir force for two identical rectangular gratings made of silicon. As shown in Fig. 1a, each grating has thickness $t$ of 2.58 μm and periodicity $p$ of 2 μm. For each rectangular protrusion, the width $w$ and length $h$ are chosen to be 908 nm and 1.5 μm respectively. Initially, the separation $s$ in the $y$-direction between the tip of the protrusions on the two gratings is 430 nm. The two gratings are offset laterally by $p/2$ such that as the bottom grating (blue) moves towards the top one (red) in the positive $y$-direction, they interpenetrate when the displacement $d$ exceeds $s$.

Calculations of the Casimir force for the full range of $d$ for this geometry, as we will later describe, requires computationally intensive numerical methods. To gain intuitive insight, we first consider a rough estimation of the Casimir force using the PFA. The analysis divides the range of displacements into 4 stages, as shown in Fig. 1b. In stage I for $d = 0$–430 nm, the PFA takes into account one plate located at the tip of a protrusion and another plate on the body of the supporting beam (e.g., the yellow lines in Fig. 1b, I). With $d < 430$ nm, the separation between these two plates that face each other in the $y$-direction is >1.5 μm, so that the total force is close to zero (black line in Fig. 1c). A sudden

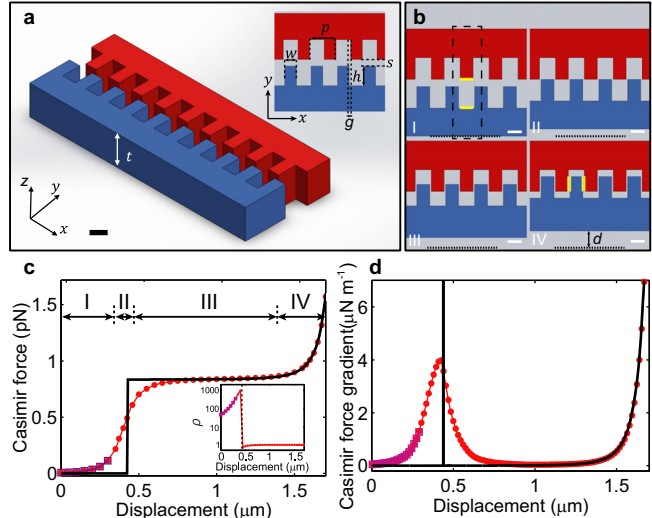

**Fig. 1 The Casimir force between perfectly rectangular gratings.**
**a** Schematic of a part of the perfectly rectangular silicon grating. Initially, the displacement $d$ of the movable grating (blue) along the $y$-direction is zero. The inset shows the top-view schematic. $h$ represents the length of a grating finger. The lateral separation between adjacent grating fingers is $g = p/2 - w \sim 92$ nm and the initial separation in $y$ is $s \sim 430$ nm. **b** Top-view schematic for the interpenetration of the two gratings. I–IV panels depict the four stages of the interpenetration. The black dotted line denotes the initial location of the bottom edge of the blue movable grating. $d$ is defined as the displacement of the movable grating from this initial position. The dashed frame encloses a unit cell. The bars measure 1 μm. **c** Calculated Casimir force per unit cell in the $y$-direction as a function of displacement $d$. The black line is the force calculated using the PFA. The red circles and purple squares are the Casimir force calculated by SCUFF-EM and the scattering theory respectively. Inset: The ratio ρ of the Casimir force to the force obtained by the PFA. The black dashed line marks where interpenetration occurs. **d** The gradient of the Casimir force.

change takes place when $d$ reaches $s$ as the sides of adjacent rectangular protrusions on the two gratings start to overlap (Fig. 1b, II). Following the common procedure of calculating the lateral Casimir force with the PFA[46], we consider the Casimir energy due to the overlap of the sidewalls that face each other in the $x$-direction (the yellow lines in Fig. 1b, IV). Since the energy increases linearly with $d$-$s$ due to the increase in the overlap area, the spatial derivative of the energy gives a constant, non-zero force that is independent of displacement. As shown in Fig. 1c, region II ($d = 430$–500 nm) contains this discontinuous jump of the force from near zero to the constant value. Contributions from the normal force between plates that face each other in the $y$-direction remain small for region III ($d = 500$ nm to 1.45 μm), so that the total force is almost constant. In region IV ($d > 1.45$ μm), the tip of the protrusions and the body of the supporting beam become close and the total force rapidly increases, in a manner similar to that between two infinite parallel plates.

In many experiments on the Casimir effect, the quantity that is directly measured is the spatial gradient of the force. Figure 1d plots the derivative of the force obtained from the PFA results in Fig. 1c as a black line. The most prominent feature is a delta function at $d = s$ when the tops of the red and blue protrusions are aligned. Other than this spike, the distance dependence of the force gradient resembles that between two parallel plates, increasing with $d$ and rising sharply when the top of the protrusions approaches the troughs on the beam. While the PFA provides a useful starting point in analyzing the Casimir force between the two perfectly rectangular gratings, the infinite force

gradient is clearly unphysical. The strong geometry dependence of the Casimir force in this system requires the use of more precise theories.

We perform numerical calculations of the Casimir force using SCUFF-EM[47], an open-source software capable of calculating the exact Casimir force between objects of arbitrary shapes provided that sufficient computation power is available. SCUFF-EM calculates the force by evaluating the integral of Casimir energy using a classical boundary elements interaction matrix (see Methods for details). The red lines from SCUFF in Fig. 1c, d show that the sharp rise in the force is smoothed out and the delta function in the force gradient becomes a finite peak. Notably, in regions III and IV after interpenetration occurs, the value of the distance-independent Casimir force given by SCUFF-EM agrees well with the PFA, while in region II, the PFA predicts an unphysical infinite force gradient.

With different parameters for the rectangular grating, the Casimir force changes but the key features in Fig. 1c, d remain. In particular, if the lateral distance $g = p/2 - w$ between protrusions on the two gratings is reduced, the distance-independent force in region III becomes significantly larger. Furthermore, the peak in the force gradient becomes higher and sharper (see Supplementary Note 2 for the calculations of Casimir force for different grating parameters).

The gratings geometry has been investigated in detail by a number of theory groups using the scattering theory. When a sufficient number of Fourier components are used, the scattering theory yields accurate calculations of the Casimir force provided that the two gratings are separated by a planar boundary. In other words, even though algorithms based on the scattering theory cannot be used to analyze the Casimir force in regions III and IV, they are applicable before the two gratings interpenetrate. In Fig. 1c, d, the results from the scattering theory are plotted as purple squares. They are calculated using the Fourier Modal Method[48] with Adaptive Spatial Resolution[49,50] (see Methods for details). At $d = 300$ nm, calculation with the number of Fourier modes equal to 100 yields 1% accuracy. In principle, the scattering theory is applicable for displacements up to $d = 430$ nm when interpenetration occurs. However, calculations beyond $d = 300$ nm are beyond our computation capability due to the computational power and time required for convergence. The good agreement between the calculations of SCUFF-EM and the scattering theory in region I provides an important consistency check on the validity of our calculations. Both calculations show strong deviations from the PFA. The deviations are plotted in the inset of Fig. 1c as the ratio of the SCUFF-EM and scattering theory calculations to the force obtained from the PFA. At $d = 430$ nm, the deviation attains maximum, reaching a factor of ~1000. The rectangular gratings can therefore generate Casimir forces with geometry dependences much stronger than previous experiments[14,37–39,46].

**Distance control by comb actuators**. Our experiment was designed to measure the Casimir force between two silicon gratings that are defined by electron-beam lithography and subsequently dry-etched into the device layer of a silicon-on-insulator wafer. The dimensions of the grating produced (Fig. 2b) is similar to the perfectly rectangular silicon gratings considered previously, albeit with the corners slightly rounded in the fabrication process. Even though the gratings are not perfectly rectangular, many of the important features of the Casimir force are retained, including the strong geometry effects and novel dependence on displacement discussed in the previous section. The measurement is performed using a monolithic platform with an integrated force gradient sensor and an actuator that controls

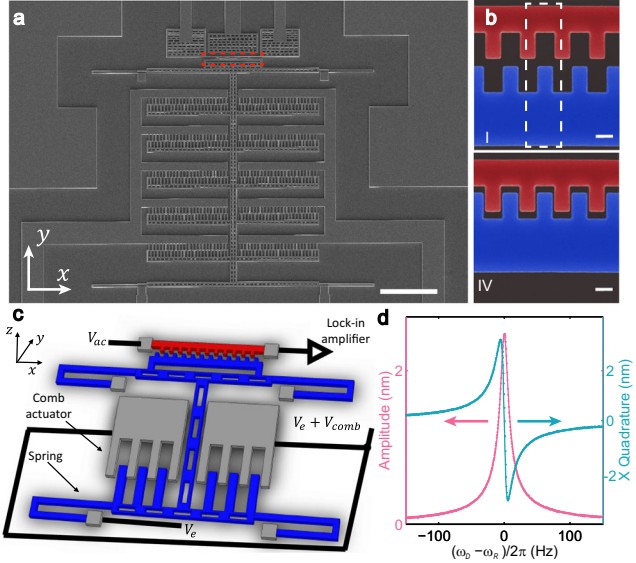

**Fig. 2 On-chip platform for force measurement and distance control.**
**a** Top-view scanning electron micrographs of the whole device. The red dash frame highlights the two sets of gratings that interact via the Casimir force. The scale bar measures 100 μm. **b** Zoom-in false-colored micrographs of a part of the gratings at displacements of 0 μm and ~1.5 μm. The white dash frame presents one unit cell of the gratings. The scale bar measures 1 μm. **c** A simplified schematic (not to scale) of the device. The gray parts, including the fixed electrodes of the comb actuator and the anchors of the movable combs, are fixed on the substrate via an underlying silicon oxide layer. The movable part of the comb actuator, colored in blue, is suspended over the substrate by four springs. The beam (red), with a length of 100 μm and a width of 1.5 μm, is excited to vibrate in-plane with amplitude of ~2 nm. It serves as a sensor for the force gradient. Gratings are attached to the beam and the moveable actuator. There are 30 unit-cells. Voltages $V_{ac}$, $V_e$, and $V_e + V_{comb}$ are applied to the beam, the movable comb and the fixed comb, respectively, as described in the main text.
**d** Mechanical response of the beam of the fundamental in-plane mode with a resonance frequency $\omega_R/2\pi \approx 1.02$ MHz and quality factor $Q \approx 91000$.

the displacement. Substantial improvements from previous experiments[14,45] are implemented to achieve the alignment accuracy and actuator stability that are essential for measuring the Casimir force between two rectangular gratings. In particular, a fabrication process involving electron-beam lithography was developed (Supplementary Note 4) to yield highly precise rectangular structures with minimal rounding of the corners.

Figure 2a shows a top-view scanning electron micrograph of the device that is fabricated using a combination of both electron-beam and optical lithography on the 2.58-μm thick device layer of a highly doped silicon-on-insulator wafer (See Methods). The red dash frame highlights the location of the gratings that consists of 30 repetitions of the unit cell depicted by the white dashed line in Fig. 2b. As shown in the schematic in Fig. 2c, one side of the gratings is located on a doubly clamped beam (red) and the other side is attached to movable comb actuators (blue). The comb actuators produce displacement in the y-direction to control the separation between the two gratings while the beam detects the force gradient exerted on the top gratings (red) by the shift in its resonance frequency.

As shown in Fig. 2a, the comb actuator consists of 10 sets of fixed and movable comb fingers. Only two sets are shown in Fig. 2c for simplicity. The gray combs are fixed to the substrate while the blue movable combs are suspended by serpentine springs. When a voltage difference $V_{comb}$ is applied between the fixed and movable combs, an attractive electrostatic force that is

proportional to $V_{comb}^2$ is generated to produce displacement of the movable comb. The blue grating is pushed towards the red one attached on the beam, with a displacement $d$ in the y-direction that is determined by the balance between the electrostatic force and the restoring force from the springs:

$$d = \alpha V_{comb}^2 \qquad (1)$$

where $\alpha$ is a proportionality constant.

Figure 2b shows the false-colored micrographs of part of the gratings. At displacements large enough for the gratings to interpenetrate, the overlapping edges of the two sets of gratings are only separated by a distance $g = p/2 - w \sim 90$ nm in the x-direction. This separation must be maintained as the lower unit is pushed towards the upper one by the comb actuator. Ideally, a perfect comb actuator produces displacement only in the y-direction. However, nonuniformities in fabrication could lead to a small, undesirable component of the displacement in the lateral (x) direction as $V_{comb}$ is applied. To meet the stringent requirement of maintaining a stable $g$, the lateral stability of our comb actuators has been improved from previous experiments by a factor of 3[14,45]. For example, serpentine springs are redesigned so that their spring constants in the x-direction exceed those in the y-direction by a factor of >100. In addition, the lateral alignment between the fixed and movable combs is also improved to minimize the difference in the distances of each comb finger to its two near neighbors. From micrographs and measurement, we estimate that $g$ changes by less than 5 nm over the full scale of displacement in the y-direction.

**Force gradient sensor and its calibration.** The grating at the top consists of rectangular protrusions from a doubly clamped beam (red in Fig. 2c) that serves as a detector of the force gradient on the grating. In the presence of a magnetic field perpendicular to the substrate, an a.c. current with frequency $\omega_D$ applied to the beam generates a periodic Lorentz force that excites the fundamental in-plane vibrational mode. As the beam vibrates in the magnetic field, a back electromotive force is induced to modify the current by an amount that is proportional to the vibration amplitude. Figure 2d shows that the vibration amplitude peaks at the resonance frequency $\omega_R/2\pi$ of ~1.02 MHz with a quality factor Q ≈ 91000. All measurements are performed at 4 K and $<1 \times 10^{-6}$ Torr.

At small separations, the grating on the movable comb (blue) exerts measurable Casimir and electrostatic forces on the grating on the beam. Due to the spring softening effect, the resonance frequency of the beam shifts by an amount $\Delta\omega_R$ that is proportional to the spatial gradient of the total force:

$$F'(d, V_e) = k\Delta\omega_R \qquad (2)$$

where $k < 0$ is a proportionality constant. The total force $F(d, V_e)$ depends on the displacement $d$ and applied voltage $V_e$ between the top and bottom gratings. It consists of two components: the electrostatic force is given by $F_e = \frac{1}{2}C'(d)(V_e - V_0)^2$ where $V_0$ is the residual voltage and $C'(d)$ is the spatial derivative of the capacitance between the two gratings in the y-direction evaluated at displacement $d$. Forces that cannot be balanced by the application of $V_e$ including the Casimir force, are represented by a second term $F_c$. Taking spatial derivative yields the gradient of the total force:

$$F'(d, V_e) = \beta(d)(V_e - V_0)^2 + F_c'(d) \qquad (3)$$

where $\beta(d) = C''(d)/2$.

Figure 3a plots the measured $\Delta\omega_R$ as a function of $V_e$ for several values of $V_{comb}$. Each $V_{comb}$ gives a fixed displacement $d$ according to Eq. (1), labeled in the figure. The contribution of the

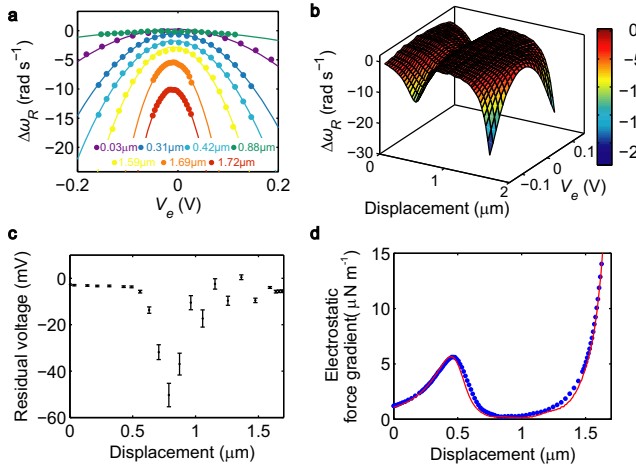

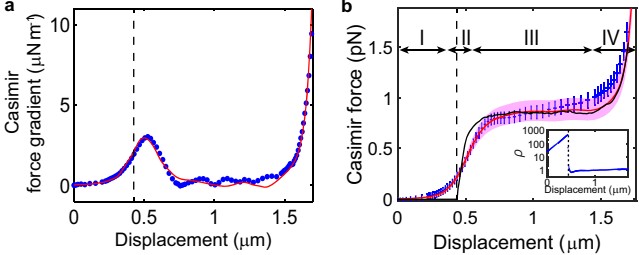

**Fig. 4 Measured Casimir force and force gradient. a** Measured Casimir force gradient (blue) as a function of displacement. The red line is calculated by SCUFF-EM based on the digitized profiles. The standard deviations over independent measurements are comparable to the dot sizes. **b** The measured force gradient is integrated over displacement to yield the force. Error bars are obtained from propagating the uncertainties in the measured force gradient. The red line is calculated by SCUFF-EM and the black line is generated by the PFA. The pink band shows the uncertainty of SCUFF-EM arising from the pixel size of micrographs (5 nm). The inset plots the ratio $\rho$ of the measured force to the force calculated with the PFA. The dashed line labels where the gratings interpenetrate.

**Fig. 3 Calibration using the electrostatic force. a** Measured $\Delta\omega_R$ as a function of $V_e$. The dots are measured data and the solid lines are parabolic fits. For each parabola, the displacement of the grating attached to the comb actuator is labeled with the corresponding color. **b** Measured $\Delta\omega_R$ as a function of displacement $d$ and $V_e$. **c** Measured residual voltage $V_0$ as a function of displacement. Error bars show the spread of data over several independent measurements. **d** Measured electrostatic force gradient with $V_e = V_0 + 100$ mV (blue) is fitted with calculations using the finite elements method (red). The error bars are comparable to the dot size.

electrostatic force gradient is quadratic in $V_e - V_0$ with coefficient $\beta(d)/k$ while voltage-independent forces including the Casimir force produce a vertical offset of the parabolas. Figure 3b shows the dependence of $\Delta\omega_R$ on $d$ and $V_e$ as a 3D surface plot. At $V_e = V_0(d)$ where each parabola attains its maximum, the contribution of the electrostatic force is minimized.

$V_o$ is measured to be close to zero, varying from 5 to 50 mV over the full range of $d$ shown in Fig. 3c. At displacements between 0.6 and 1.4 μm, the electrostatic force gradient is close to zero. The parabolic fits have small curvatures (e.g., the one at 0.88 μm in Fig. 3a) and give large uncertainties in $V_0$. Furthermore, the gratings under measurement are not perfectly rectangular in the top view and the sidewalls are not perfectly smooth. Different parts of the interacting surfaces have different crystal orientations, especially the rounded corners. Variations in the work function[51] could result in $V_0$ not constant with displacement.

The constants $\alpha$ for the comb actuator and $k$ for the sensing beam are calibrated by fitting the frequency shift induced by the electrostatic force gradient to $\beta(d)$ calculated using finite-element simulations by the numerical package COMSOL. As discussed in Methods, the boundary conditions used in the calculations are obtained from the digitized top views of the sample. Figure 3d plots a least-square fit of the measured electrostatic force gradient per unit cell at $V_e - V_0 = 100$ mV, yielding $\alpha = 1.05 \times 10^{-6} \pm 2.15 \times 10^{-8}$ N m$^{-1}$s rad$^{-1}$ and $k = -8.73 \pm 0.03$ nm V$^{-2}$. The fitting process scales $V_{comb}^2$ and $\Delta\omega_R$ in the experiment (blue dots) by factors $\alpha$ and $k$, respectively, to minimize deviations from the calculated electrostatic force as a function of displacement (red line). There is good agreement between measurement and the fit.

**Comparison of measured force gradient with theory**. We minimize the contributions of the electrostatic force by setting $V_e = V_0$ and measure $\Delta\omega_R$ as $V_{comb}$ is increased. Using the calibrated values of constants $\alpha$ and $k$, the results are converted into the dependence of the force gradient on $d$, as shown by the blue data in Fig. 4a. The force gradient is then integrated over displacement

to yield the force as a function of $d$ in Fig. 4b. The uncertainty of force accumulates during the integration leading to error bars increasing with displacement. Calculations of the Casimir force and force gradient for gratings of the same shape as those in our experiment are plotted as red lines. The calculations are performed with SCUFF-EM, using a geometry obtained from digitizing the top-view scanning electron micrograph of the gratings (see Supplementary Note 1 for the digitizing micrographs). Each unit cell is assumed to be infinite and invariant in the z-direction. Calculations of the Casimir force is repeated for six different grating units along the beam to yield an averaged value. The finite conductivity of silicon is included (see Methods). To simplify the calculations, a temperature of 0 K is used instead of the actual temperature of 4 K in the experiment. This approximation is justified because the separation between the relevant parts of the two bodies is smaller than 500 nm for all displacements. For example, at displacement $d = 1.6$ μm where the top of the grating is about 0.3 μm from the main body of the beam, the calculated Casimir forces for 0 K and 4 K differ by <0.3%.

The measured force/force gradient on the gratings is in good agreement with the SCUFF-EM calculations. In particular, the peak in the Casimir force gradient at the onset of interpenetration ($d \sim 0.43$ μm, labeled by the dashed line) and the sharp rise when the tip of the grating approaches the main body of the beam (stage IV) are both reproduced in the measurement. The four regions discussed in the section for perfect rectangular gratings can be readily identified in Fig. 4b. Specifically, region I ($d = 0$–430 nm) corresponds to the range of displacement before interpenetration. The measured force rapidly increases when interpenetration occurs in region II (430–600 nm). In region III ($d = 600$–1450 nm) the force is nearly independent of displacement. The force increases rapidly in region IV ($d > 1450$ nm) due to the interactions between the top of the gratings and the body of the beam. However, there are also a number of important differences from perfect rectangular gratings. First, the force gradient in Fig. 4a peaks at a displacement that is larger than the onset of interpenetration (marked by the dashed line in Fig. 4a), instead of before the onset as in Fig. 1d for rectangular gratings. Such difference can be attributed to the rounded corners of the grating fingers. Second, the measured force gradient shows small fluctuations about zero for $d$ between 0.8 and 1.4 μm due to the roughness in the sidewalls. These fluctuations are also present in the calculations that are based on the top view of six units. From

images of the sidewall, we estimate the roughness to be less than 10 nm. In Fig. 4b, the measured force shows a slight increase with displacement in region III rather than remaining constant as in the SCUFF-EM calculations. This increase is attributed to a non-zero mean force gradient that accumulates in the integration to yield the force. Plausible reasons for the deviation include effects due to patch potentials[51], lateral shift of the movable combs not accounted for in our model, sidewall imperfection and uncertainty in digitizing the micrographs. For example, we estimate the uncertainty introduced to the Casimir force by expanding and shrinking the digitized boundary by 2.5 nm (within one-pixel of micrographs) in the normal direction of the boundary. The results are shown as the pink band in Fig. 4b. Taking into account the sensitive dependence of the force on the separation of surfaces and the difficulties in accurately determining the shape of the structure, we consider the measured force in good agreement with theoretical calculations of the Casimir force.

As discussed previously, the Casimir force on perfect rectangular gratings shows a strong geometry dependence. Due to the rounded corners in our samples, the geometry dependence is slightly weakened in our samples. Nevertheless, the measured deviation from the PFA strongly exceeds those from previous experiments[14,37–39,46]. The black line in Fig. 4b shows the force produced by the PFA on the digitized geometry of the grating. It is near zero in region I and increases sharply in region II once interpenetration of the two gratings takes place. In regions I and II, results from the PFA deviate from both the measured force and the force calculated using SCUFF-EM. In particular, the smooth increase of the measured Casimir force becomes an abrupt change in the PFA. In the inset of Fig. 4b the ratio between the measured force and the PFA shows a peak at a value of ≈500 at $d \approx 426$ nm, which is weaker than the 1000 times deviation shown in the perfectly rectangular silicon gratings (Fig. 1c). Nevertheless, such geometry dependence is stronger than those observed in the grating-plane geometry[38,39] by a factor >100.

In region III, after the two gratings interpenetrate, the Casimir force is non-zero but is almost independent of distance. The force is expected to depend strongly on the lateral distance $g$ between two adjacent grating fingers. While it is not feasible for us to fabricate many different devices with different $g$ to study this behavior, we analyze the dependence of the force on $g$ using the PFA as it agrees well with the exact calculations of SCUFF-EM in region III. So far, all the results of the PFA presented are based on the real optical properties of silicon used in the experiment. For simplicity and without loss of generality, we now consider rectangular gratings made of perfect metal separated by different values of $g$. The PFA considers the overlapping surfaces as parallel plates with energy $\pi^2 \hbar c / 720 g^3$ per unit area. As the overlap area increases linearly with displacement, the Casimir force given by the spatial derivative of energy remains constant, with a magnitude inversely proportional to $g^3$. A similar study is also performed to determine the dependence of the peak in the force gradient (Fig. 4a) on $g$, as described in Supplementary Note 2. For gratings made of materials with finite conductivity, it is expected that for small $g$ the scaling will change to $1/g^2$ in the nonretarded limit.

## Comparisons to pairwise-additive approximation.

Apart from the PFA, another well-known method to estimate the Casimir force is the pairwise additive approximation (PAA)[52,53]. It divides the interacting objects into elementary constituents and sums up the interaction energy under the assumption that the interaction between two elements is not affected by the presence of others. We calculate the force using PAA (see Methods) and compare it to the exact Casimir force calculated by SCUFF-EM in Fig. 5a, b for the

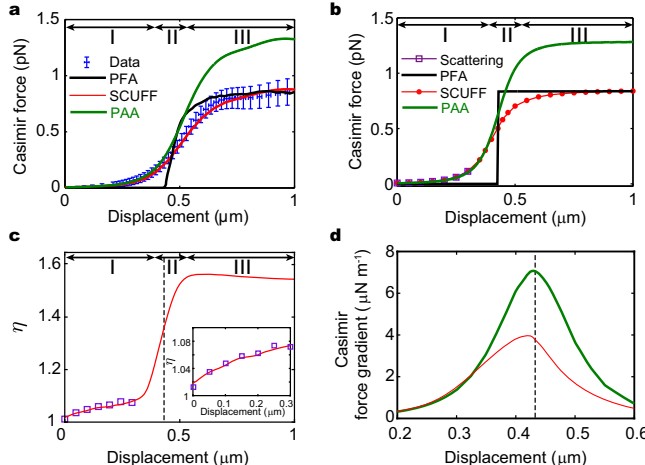

**Fig. 5 Force calculated with the pairwise additive approximation.** Comparison of the Casimir force calculated by SCUFF-EM (red), PAA (green), measurement (blue), and PFA (black) for one unit cell of the silicon grating that is **a** digitized from the top view of the experimental device and **b** perfectly rectangular. The measured data and error bars are the same as Fig. 4. **c** Ratio η of the forces calculated by PAA to scattering theory (purple) and SCUFF-EM (red) for perfectly rectangular gratings. The black dashed line marks where interpenetration occurs. Inset: Zoom-in for displacement between 0 nm and 300 nm. **d** The gradient of the Casimir force calculated using PAA (green) is symmetric about the dashed line that marks the distance at which interpenetration takes place for silicon gratings that are perfectly rectangular. The actual Casimir force gradient peak (red) is at a slightly smaller displacement and is asymmetric. The optical properties of silicon are used for all calculations (see Methods).

silicon grating geometry of our device and the silicon gratings that are perfectly rectangular considered at the beginning of the paper, respectively. In most of region I, before interpenetration, the PAA provides a good estimate of the Casimir force. However, in regions II and III deviations become apparent. In particular, in region III the PAA reproduces a non-zero force that is largely independent of $d$. However, the magnitude is overestimated by ~50% (Fig. 5c). The overestimation represents a breakdown of the PAA since it regards the medium between the interacting elements as vacuum. While the determination of the applicable range of the PAA is beyond the scope of this paper, the observed behavior appears to be consistent with the notion that the PAA generally works better when the separation between the interacting bodies is large, so that the vacuum media assumption is nearly valid[54]. For example, in region I the two gratings are far from each other while in region III, the lateral separation $g$ between adjacent grating fingers is only ~90 nm. In the regime of interpenetration, it is more likely for material to be present between two interacting elements. Intuitively, it is expected that deviations from PAA are larger in region III compared to region I.

The inset of Fig. 5c plots the ratio of the forces calculated by the PAA to that by the scattering theory and SCUFF-EM for silicon gratings that are perfectly rectangular, in purple and red, respectively. As expected, the purple and red results largely coincide with each other because both the scattering theory and SCUFF-EM work well in Region I before interpenetration (Fig. 1). At displacement of 0.3 μm, the ratio is ~1.07. The value decreases as the two gratings move farther apart.

Figure 5d plots the peak in the force gradient of the perfect rectangular gratings calculated by PAA in green. Unlike the exact Casimir force gradient calculated by SCUFF-EM (red), the peak from PAA is symmetric about the displacement where the interpenetration of the two gratings occurs. For the perfect

rectangular grating, the asymmetry of the peak in the force gradient is therefore indicative of the breakdown of the PAA.

## Discussion

It is instructive to compare the Casimir forces in our interpenetrating gratings to 3D sealed cavities[55–58]. The latter includes 3D pistons where one of the plates is movable. It has been predicted that interesting effects such as repulsive Casimir forces occur in these geometries. While the shape of our device in the regime of interpenetration (Fig. 2b, III) bears some resemblance to 3D sealed cavities, there are important fundamental differences. First, our gratings only confine the electromagnetic fields in the $x$–$y$ plane. There is no confinement at all in the $z$-direction normal to the substrate. Second, the presence of the lateral gap between the fixed and movable gratings makes the boundary conditions completely different from sealed cavities where such gaps are absent. Therefore, we do not anticipate that our devices can yield insights on Casimir effects in sealed cavities.

In conclusion, using an integrated on-chip platform, we measure the Casimir force between two nanoscale rectangular silicon gratings that are accurately aligned so that they interpenetrate as the distance between them is reduced using a comb actuator. Right before interpenetration occurs, the measured Casimir force shows a geometry dependence that is much stronger than previous experiments. To verify the validity of our calculations of the Casimir force using boundary element methods, we compare the results of a perfect rectangular grating to scattering theory and obtain good agreement. As the gratings interpenetrate each other, a novel distance dependence of the Casimir force emerges. The measured force is largely independent of displacement, with a non-zero magnitude determined by the lateral separation between adjacent grating fingers. Estimations of the force by the PFA and the PAA yield different values. The PAA works well only for the region before interpenetration while the PFA reproduces the non-zero displacement independent force after the two gratings interpenetrate. Our work opens opportunities to design structures to yield Casimir forces that strongly exceed the PFA. The possibility to align nanoscale features on two objects with high accuracy paves the way for investigating Casimir physics in novel and complex geometries.

## Methods

**Device fabrication**. The devices are fabricated on a boron p-doped silicon-on-isolator wafer with 2.58 μm device layer and 2 μm buried oxide layer [see Supplementary Note 3 for the details of the fabrication process]. Using the Van der Pauw method, the sheet resistance of the device layer at 4 K is measured to be 0.013 Ω cm, corresponding to carrier concentration of $6.0 \times 10^{18}$ cm$^{-3}$.

The two gratings are defined by a single electron-beam lithography step, which ensures they are accurately aligned. Other larger structures including the comb actuator and the serpentine springs are defined by optical lithography to reduce the fabrication time. The patterns of photo- and electron-beam resist are transferred onto a polysilicon-stacked-on-silicon-oxide etch mask. Two layers of the mask are utilized to improve the accuracy of the defined pattern. Without the protection of the etching mask, the exposed silicon is removed by the deep reactive ion etch. Next, the movable electrode and the beam are freed by etching away the buried oxide layer under it with hydrofluoric acid. The etching time is controlled so that the anchors of the four springs remain fixed on the handle wafer to support the suspended movable comb.

The hydrofluoric acid also removes the native oxide and passivates the silicon to prevent the formation of native oxide for several hours. We put the sample into a sealed probe within this time window, and pump the chamber to pressure $\sim 1.0 \times 10^{-6}$ Torr. After that, we load the sample into 4 K liquid helium.

**Calculations of electrostatic force and the Casimir force**. The calculation of the electrostatic/Casimir force is based on one unit cell of the gratings digitized from the scanning electron micrograph of the top of the structure [See Supplementary Note 1]. We reduce the calculation of the electrostatic/Casimir force into a two-dimensional problem where the shape in the $z$-direction is assumed to be invariant and infinite. The effects of the substrate are negligible because the distance between the structures is around 90 nm (Fig. 2b III, IV) which is much smaller than the distance of 2 μm from the substrate.

The electrostatic force is calculated using COMSOL. For calculations of the Casimir force, the dielectric function of silicon $\varepsilon(i\xi)$ is given by[59]:

$$\varepsilon(i\xi) = 1.035 + \frac{(11.87 - 1.035)}{(1 + \xi^2/\omega_0^2)} + \omega_p^2/[\xi(\xi + \Gamma)], \quad (4)$$

where $\omega_0 = 6.6 \times 10^{15}$ rad s$^{-1}$, $\omega_p = 2.37 \times 10^{14}$ rad s$^{-1}$, $\Gamma = 6.45 \times 10^{13}$ rad s$^{-1}$. The expression is based on the Lorentz–Drude model where the first two terms describe the dielectric function of intrinsic silicon[60]. The last term accounts for the extra carriers due to doping[61] where $\omega_p$ and $\Gamma$ are deduced from the measured sheet resistance of the doped silicon (0.013 Ωcm). An effective mass of 0.34 $m_e$ is used for electrons in the p-doped silicon.

Calculations with the scattering approach were performed using the theoretical framework described in refs. [62,63]. This method is based on a plane-wave description of the electromagnetic field in any region of space, while the bodies involved (of arbitrary geometry and material properties) are described in terms of their classical scattering (reflection and transmission) operators. This framework has been more recently applied to study the Casimir force[32,33] and the heat transfer[50] between gratings. In this case, the scattering operators have been obtained by using the Fourier Modal Method[48], based on a Fourier decomposition of the field explicitly taking into account the periodicity of the system and the introduction of the number of Fourier components of the field as a convergence parameter (see ref. [32] for details). More specifically, we have employed Adaptive Spatial Resolution[49], a modification introduced to accelerate convergence, in particular in the case of metals.

**Force calculations using the PAA**. The full van der Waals potential energy between two identical atoms with polarizability $\alpha(\omega)$, separated by a distance $r$ is given by[64]:

$$U_{A-A}(r) = -\frac{\hbar}{\pi} \int_0^\infty d\xi \frac{\xi^4}{c^4} \frac{\alpha^2(i\xi)}{(4\pi\varepsilon_0)^2} \left[ \frac{3c^4}{\xi^4 r^4} + \frac{6c^3}{\xi^3 r^3} + \frac{5c^2}{\xi^2 r^2} + \frac{2c}{\xi r} + 1 \right] \frac{e^{-\frac{2\xi r}{c}}}{r^2} \quad (5)$$

where $\omega = i\xi$ is the imaginary frequency, $\alpha$ is polarizability of the atoms, $\varepsilon_0$ is the permittivity of vacuum, and $c$ is the speed of light. For the pairwise summation method, the potential between each atom in the first object with each atom in the second object is summed. The summation is performed by integrating $U_{A-A}$ over the volumes of the objects $V_A$ and $V_B$ weighted by the number density $N$ of atoms:

$$U_c = -\frac{\hbar}{\pi} N^2 \int_{V_A} d^3 r_A \int_{V_B} d^3 r_B \int_0^\infty d\xi \frac{\alpha^2(i\xi)}{(4\pi\varepsilon_0)^2} \left[ \frac{3}{r^6} + \left( \frac{\xi}{c} \right) \frac{6}{r^5} + \left( \frac{\xi}{c} \right)^2 \frac{5}{r^4} \right.$$
$$\left. + \left( \frac{\xi}{c} \right)^3 \frac{2}{r^3} + \left( \frac{\xi}{c} \right)^4 \frac{1}{r^2} \right] e^{-\frac{2\xi r}{c}} \quad (6)$$

The polarizabilities can be replaced by the dielectric function using the Clausius–Mossotti relation:

$$\frac{\varepsilon(\omega) - 1}{\varepsilon(\omega) + 2} = \frac{N\alpha}{3\varepsilon_0} \quad (7)$$

yielding:

$$U_c = -\frac{\hbar}{\pi} \left( \frac{3}{4\pi} \right)^2 \int_{V_A} d^3 r_A \int_{V_B} d^3 r_B \int_0^\infty d\xi \left( \frac{\varepsilon(i\xi) - 1}{\varepsilon(i\xi) + 2} \right)^2$$
$$\left[ \frac{3}{r^6} + \left( \frac{\xi}{c} \right) \frac{6}{r^5} + \left( \frac{\xi}{c} \right)^2 \frac{5}{r^4} + \left( \frac{\xi}{c} \right)^3 \frac{2}{r^3} + \left( \frac{\xi}{c} \right)^4 \frac{1}{r^2} \right] e^{-\frac{2\xi r}{c}} \quad (8)$$

Details of the algorithm for calculation of Eq. (8) for our geometry are presented in Supplementary Note 5. The forces and force gradients in Fig. 5 are obtained by taking the first and second spatial derivatives of $U_c$.

## Data availability

All data supporting the findings of this study are available within the article and its Supplementary Information or from the corresponding author upon reasonable request.

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

## Acknowledgements

M.W., L.T., C.Y.N., C.T.C., and H.B.C. are supported by AoE/P-02/12 from the Research Grants Council of Hong Kong SAR. M.W., L.T., and H.B.C. are also supported by HKUST 16300414 from the Research Grants Council of Hong Kong SAR. Research by M.A., B.G., and R.M. was carried out using computational resources of the group Theory of Light-Matter and Quantum Phenomena of the Laboratoire Charles Coulomb.

## Author contributions

H.B.C. conceived the idea of the work. M.W. and L.T. performed the experiments and analyzed the data. M.W., C.Y.N., and C.T.C. carried out calculations using SCUFF-EM. R.M., B.G., and M.A. performed calculations using scattering theory. J.A.C. and M.W. calculated the force using PAA. M.W. and H.B.C. co-wrote the paper.

## Competing interests

The authors declare no competing interests.
