## [Peer Review File · Nature Communications]

REVIEWER COMMENTS

Reviewer #1 (Remarks to the Author):

In this manuscript, the authors describe their measurements of the Casimir force between two interpenetrating gratings using a MEMS actuator. They find significant deviations from the PFA and PAA calculations (as expected) and show good agreement with calculations using SCUFF-EM. In general, the findings are well-presented and will likely be of interest to both theorist and experimentalist studying the Casimir effect. Below are my questions for the authors.

- 1) In the abstract the authors state that they “measure the Casimir force between two rectangular gratings in regimes not accessible before.” They later describe in the text how the measurements presented in this paper are improved over those from their recent Nature Photonics paper (ref 14), where they observed non-monotonic Casimir forces in a slightly more complex geometry. Was that improvement really necessary to see this effect? I think that the manuscript could benefit from a little more discussion putting this measurement in context with their previous measurement, which seemed similar but slightly more complex. I believe the main advantage of this geometry is the larger discrepancy between experiment and PFA/PAA, but the authors should clarify.
- 2) Related to the PFA discrepancy, the authors write that it is either a factor of 500 or 1000 at different places in the text. This should be clarified as well.
- 3) On page 3, there is a discussion of the effect of geometry on the Casimir effect, noting that most experiments are performed in the sphere-plate configuration. The authors should also note that experiments have been performed in the plate-plate (Bressi, Phys. Rev. Lett. 88, 041804, 2002) and sphere-sphere (Garrett, Phys. Rev. Lett. 120, 040401, 2018) configurations.
- 4) Figure 1: For the inset in (a), “g” is not clear. Also, for (c) and (d) and elsewhere in the manuscript, “displacement” is used to describe the separation between parts of the structure. The authors should clearly identify what is mentioned by “displacement” by assigning it a variable and putting it in the inset of (a) or by using a different variable for the x-axis, e.g. “s” which appears in the inset.
- 5) For the PFA, it appears that the authors are assuming ideal metal plates rather than Si, which they use for the PAA. Is that the main difference? I don’t think so, but the authors should explain a little more the difference between these two calculations because it appears that they are different approximations about both geometry and optical properties. This will be helpful to make the conclusions of greater interest to the broader readership.

Reviewer #2 (Remarks to the Author):

The paper “Strong geometry dependence of the Casimir force between interpenetrated rectangular gratings” presents new experimental measurements of the Casimir force in the system of two aligned rectangular gratings, comparison with the theory is performed. Results of the paper should be interesting to a general reader and specialists in the field. In my opinion, the paper may be suitable for publication in Nature Communications after authors clarify subtle points and consider recommendations to improve the paper written below.

In the paper under consideration the Casimir effect in the system of two aligned rectangular gratings is studied. For this system the derivative of the Casimir force versus separation between two rectangular gratings is measured in experiment, the comparison between theory and experiment is performed. Authors study both the case when the two gratings do not penetrate into each other and the case when the two gratings interpenetrate each other. The regime of interpenetration was never studied in Casimir experiments before. Authors were able to perform comparison of several exact and approximate theoretical methods with experimental results for the system of two rectangular gratings

at various separations between them. This is the first Casimir effect experiment for two rectangular gratings separated by a vacuum slit, the experiment is in agreement with the theory. Note that in most previous experiments Casimir measurements were performed in a system of a sphere and another geometry separated by a vacuum slit. Here two gratings separated by a vacuum slit were aligned with high accuracy, this technique may be used in future for direct comparison of the Casimir theory and experiments in various geometries different from a sphere.

I do have several questions and recommendations to improve the paper written below.

1. Theory for evaluation of the Casimir energy between two aligned gratings in terms of Rayleigh reflection coefficients was developed in the paper [27] and in the paper with a lateral displacement of gratings:

A.Lambrecht and V.N.Marachevsky, *Int.J.Mod.Phys.A* 24, 1789 (2009).

This fact and these references should be mentioned since authors obviously use general expressions for the Casimir force in terms of Rayleigh coefficients in their calculations shown by purple squares on Fig.1 and Fig.5.

2. What is the temperature T in the experiment (I could not find the value in the text) ? Why is it possible to use zero temperature calculations in comparison of the theory and experiment ? If authors use the last term in the righthandsight of the formula (4) at finite temperature (a Drude term), is there an agreement between the theory and experiment (did they check this) ? What is the difference between zero temperature and finite temperature results for the force ?

It would be really good to comment on this in detail. Also it would be good to draw a plot with the ratio of zero and finite temperature theoretical results for the Casimir force at various separations.

3. Authors should define the regions I, II, III, IV in nanometers and should emphasize how these regions are defined in general so that the readers could fully understand the boundaries between these regions. It is unclear to me where the regions II, III, IV start.

4. On line 189 it is written $\sim 1000\%$, however, in all other places it is written $\sim 500\%$. Probably there should be $\sim 500\%$ on line 189 as well.

5. It would be good to specify explicitly in the text the smallest distance between rectangular gratings for which authors were able to perform calculations within a scattering theory.

6. It would be good to discuss in more detail the difference between PAA and results of scattering theory. I expect it would be really interesting to add the plot where the ratio of PAA force to the exact force is explicitly shown at various separations.

7. While discussing pistons in Discussion section I recommend the reference:

V.N.Marachevsky, *Phys.Rev.D* 75, 085019 (2007).

In this paper exact formulas for a piston with an arbitrary cross section were derived.

8. In Supplementary information:

line 13 – should probably be “1 nm”,

line 17 – should probably be “As discussed in the main text, ...”.

Reviewer #3 (Remarks to the Author):

Wang, Tang, Ng, Messina, Guizal, Crosse, Antezza, Chan, and Chan
Strong geometry dependence of the Casimir force between interpenetrated rectangular gratings
submitted to Nature Communications
ms# 20-34993

The authors present an experiment that measures the van der Waals-Casimir force between two nanofabricated gratings of nearly rectangular shape that approach each other until they interpenetrate. The force gradient is measured by the frequency shift of a transverse oscillation of one of the gratings, and compared to numerically exact and approximate calculations. A good agreement with a fully numerical result (based on the code `scuff-em`) is found. Other approximations (like proximity force PFA or pairwise additivity PAA) fail in one or the other region of distances, for which the authors give heuristic explanations.

These results improve on the data shown in Ref. 14 by a team around H. B. Chan: the fabrication of the gratings is much better so that rectangular shapes are very closely achieved. The theory team has changed and is able to contribute calculations with different methods and approximations, although the full numerics is based on the same `scuff-em` code.

So why should this represent a significant advance to be published in Nature Communications?

A similar good agreement between experiment and theory was presented by Zou & al (Ref. 42) and Tang & al (Ref. 14).

The fact that interpenetration leads to a constant Casimir force has already been mentioned and interpreted in earlier work, too, in particular in Refs. 14, 44.

The deviation with respect to PFA is exacerbated here (a relative factor of a few hundred), but it seems that this is due to a kind of simplistic way to apply the PFA, in particular when one deals with rectangles or rounded corners. In Ref. 14, the rounded corners of the T-shaped structures were taken into account with PFA (but perhaps it was rather the PAA that was applied there), giving a good qualitative agreement with `scuff-em`. So one may object that the "selling point of ~ 500 " is an artefact of an oversimplified PFA. And since it is quite obvious that the PFA is not applicable for this kind of system and since numerically exact calculations are available, it does not seem convincing to build the significance of the paper on the failure of an "out-dated approximation".

But this is perhaps the main insight of this material: one can now fabricate structures where the PFA is utterly wrong (in some parameter region, as edges/corners slide against each other), and still one can demonstrate really good agreement between theory and experiment.

The authors should try to put their work into perspective in view of these remarks.

A few minor points are listed below. It may also be worth considering a shorter presentation of the experimental apparatus, since the same has been used in the Refs. mentioned above. The reader would rather learn more details about how the nearly perfect rectangular shapes have been manufactured: which processing steps led to success?

If these questions are answered, then indeed, this makes an interesting paper that demonstrate that a significant challenge in the field of dispersion forces has been met. Worth being published in Nature Commun.

Carsten Henkel

...

p2, line 61-62: I think that Ref. 24 should not be over-advertised here. The heat transfer is actually ridiculously small, since it concerns just single modes in two membranes. And there are many other examples where the van der Waals force has mechanical consequences, as in the classical analysis of thin liquid Helium films.

line 67: "pair of plates" > "pairs of plates"

p4, line 99: the regime of interpenetration has been explored, indeed, in Ref. 14 (Tang & al.), just with a slightly different shape of the protrusions.

line 139: the "d" in the formula should be in italics

line 165: Casimir "path integral" -- I know that there are path integral formulations for the Casimir energy, but in the context of numerics like scuff-em, I don't think that this is an efficient starting point. Perhaps you mean "surface integral" which is taken, in this effective 2D geometry for the numerics, along a path, indeed. But the word "path integral" is fixed to Feynman's integral, please avoid the confusion.

line 170: the sentence on the breakdown of PFA is a bit too much, that has been said a few lines before.

line 184: what is "N" in " $N = 100$ "? the number of Fourier modes? then just say so and drop the symbol.

Fig. 2(b): why repeat the sketches / micrographs with the interpenetrating fingers from Fig. 1? I know it's not a sketch, but it's still graphically a redundant information.

caption Fig. 2, line 216: drop " $\omega_R =$ " or add the 2π that is needed here. The number of digits for the quality factor Q is ridiculous, I don't believe you can be so precise (same problem in the main text).

p10, around line 220: I would like to know the number for the thickness of the facing gratings (along the z-direction).

A simple basic question: it seems that the top beam (in red in Fig. 2) is "stiffer" than the lower grating (blue). Does that not mean that as the top beam is oscillating, it will "entrain" (via the Casimir force) the lower one. How can you be sure that the lower grating is kept in place and is not oscillating, too? Perhaps it is just a question of mass (the lower structure is much larger and more massive)?

line 261: correct spelling is "Torr".

line 308: replace V_o by V_0 (index zero)

p15, lines 319-21: "Thermal corrections are neglected as the zeroth Matsubara frequency term accounts for nearly all of the force" -- something is misunderstood here. The zero'th Matsubara term gives the *high-temperature limit* of the dispersion energy. You rather mean that the *integral* over

imaginary frequencies which deviates from the sum (i.e., the force at 4K) only by less than 0.3%. Or did I misunderstand something here?

p17, around line 363: the abrupt change in the PFA result seems like an artefact of having sharp corners (rectangular profile). If these corners are rounded (as shown by the TEM scan), then also the PFA can be modified to give a smooth result. This has been done in Ref. 14, for example. For sure, the discrepancy between experiment and "this version of the PFA" will be smaller, as also shown in Fig. 1 of Ref. 14.

line 365: the peak value 473 is too precise, I would only bet on something rounded like 500. You should be aware of your experimental errors for that ratio ...

p17, line 374: the universal Casimir formula $\hbar c \dots / g^3$ is misleading here. As mentioned in my comments on the Supp Mat, one expects at the small distances here, that the energy per area rather follows the conventional Hamaker (nonretarded) formula A / g^2 (with material-specific Hamaker constant A). There is a problem with units in the Supp Mat calculation that leads to $1/g^3$.

p18, around line 383: you mention in the details in Methods that the van der Waals potential used in the PAA assumes that there is no material between any two surface (even volume) elements. So it is intuitive to understand that this does not apply for the corners of the fingers when they interpenetrate, because they interact mainly across silicon (fingers in).

line 390-91: the sentence "failure of PAA is from the non-pairwise additive nature" is redundant (french: pleónasme) and does not say anything.

Fig. 5(a): I again would like to understand why the PFA gives a sharp onset, while in Fig. 1 of Ref. 14, the PFA starts off smoothly. Do your structures have so much sharper corners here?

line 430: replace "As the displacement is further increased so that the gratings interpenetrate each other" by "As the gratings interpenetrate each other" for those who only rapidly read the Conclusion.

line 437: typo "paths" > "paves (the way)"

line 456: I would expect "load the sample" rather than "load the probe"

line 469: I rapidly computed the plasma frequency for the given density of p-carriers and found a slightly different value, although I took the given effective mass:

```
>>> hole_n = 7.2e18/cm**3
>>> omega_p = sqrt(hole_n*e0**2/eps0/(0.34*me))
>>> omega_p/1e14
2.60
```

Or does one need the effective mass of holes rather than electrons?

Methods, Eq.(8): well, this sounds like a horrible integral. Is there no way to find a reasonable approximation here? For the typical ξ 's in $\epsilon(\xi)$, unfortunately $c / \xi \sim 300$ nm, comparable to typical distances, so you may not rely on simple power law approximations. But there must be reasonable Padé approximations for that, no?

References: the citation style is not uniform (journal names in full or abbreviated, lower case or upper

case). A number of references are incomplete:

Ref. 31 -- Phys. Rev. A 86 (2012) 062502
Ref. 35 -- Phys. Rev. Lett. 105 (2010) 250402
Ref. 36 -- Phys. Rev. Lett. 101 (2008) 030401
Ref. 37 -- Nature Commun. 4 (2013) 2515
Ref. 38 -- Phys. Rev. A 82 (2010) 062111
Ref. 44 -- Phys. Rev. B 81 (2010) 115417
Ref. 47 -- twice J. Opt. Soc. Am. A
Ref. 48 -- Phys. Rev. B 95 (2017) 125404
Ref. 50 -- Phys. Rev. Lett. 118 (2017) 266802
Ref. 53 -- Phys. Rev. D 69 (2004) 065015

Supp Mat

p3, line 25: try to improve the ordering of the information. Suggestion: "between two ... corners, see inset in (a): two blocks initially separated by a gap of 80 nm (x-direction) and 500 nm apart in the y-direction. Main figures (a) and (b): calculated Casimir force and force gradient as ...

p3, line 32: if your bodies are infinite in the z-direction, you can only compute forces per unit length along that direction. So the units in Fig. 2 are wrong. Only the force per unit area, between perfect conductors, can scale like $1/g^3$, with the same exponent as the energy per area for two infinite plates (linear increase with the area as the bodies slide one against the other).

p4, line 42: I think that there is no "consistency with the simple PFA argument" of the main text. (1) In the experiment, no exponent in the gap distance g can be found. (2) At the ~ 70 nm distance, it is likely that the Casimir energy in the experiment is already in the non-retarded (or Hamaker) regime, while for perfect reflectors, there is no non-retarded regime. Hence, the energy/area (and the force per unit length) scales like $1/g^2$. Since the length along the z-direction is constant, this scaling should also apply to the experimental force.

Response to Reviewer 1

We are pleased that Reviewer 1 thinks that our paper “will likely be of interested to both theorist and experimentalist studying the Casimir effect” and “are well-presented”. We are grateful to her/him for the suggestions for improvements and positive comments. We respond to her/his suggestions below:

In the abstract the authors state that they “measure the Casimir force between two rectangular gratings in regimes not accessible before.” They later describe in the text how the measurements presented in this paper are improved over those from their recent Nature Photonics paper (ref 14), where they observed non-monotonic Casimir forces in a slightly more complex geometry. Was that improvement really necessary to see this effect? I think that the manuscript could benefit from a little more discussion putting this measurement in context with their previous measurement, which seemed similar but slightly more complex. I believe the main advantage of this geometry is the larger discrepancy between experiment and PFA/PAA, but the authors should clarify.

The improvements in the fabrications are indeed essential in revealing the strong deviations of the Casimir force from PFA and PAA. The main difference is that we changed from optical lithography to electron beam lithography so that the sharp corners with little rounding can be achieved and the uniformity among individual units is significantly improved. The non-monotonic behavior of the Casimir force demonstrated in the previous experiment [14] does not require the corners of the structures to be sharp. However, the strong deviation from PFA in the current paper does. In Ref. 14 with rounded T-protrusions, the deviations of the Casimir force from the PFA is only $\sim 40\%$, much weaker than the factor of 500 demonstrated in this paper.

We added Supplementary Notes 3 to describe the new fabrication process that uses electron beam lithograph and Supplementary Notes 4 to compare the two experiments. We also added the following sentences to the main text:

“A prior experiment measured the non-monotonic Casimir force when two T-shaped protrusions interpenetrate¹⁴. However, due to the limited resolution of optical lithography in the fabrication process, the protrusions are rounded at the corners. Moreover, there are non-uniform among the different units, introducing uncertainties so that deviations from the PFA cannot be unambiguously identified. To our knowledge, the strong geometry dependence of the Casimir force in the regime of interpenetration for rectangular gratings remains unexplored.”

“In particular, a fabrication process involving electron beam lithography was developed (Supplementary Note 4) to yield highly precise rectangular structures with minimal rounding of the corners.”

Related to the PFA discrepancy, the authors write that it is either a factor of 500 or 1000 at different places in the text. This should be clarified as well.

The different factors correspond to two different geometries. For the rectangular silicon gratings with perfectly sharp corners, the deviation can reach 1000 times based on the numerical simulation from SCUFF and the semi-analytical scattering theory. For the device that is measured, the deviation is ~ 500 due to slight rounding of the corners in the fabrication process. In the abstract and conclusion, we use the experimentally demonstrated deviation of 500 times.

We have added the following text to clarify:

“In the inset of Fig. 4b the ratio between the measured force and the PFA shows a peak at a value of ≈ 500 at $d \approx 426$ nm which is weaker than the ≈ 1000 times deviation shown in the perfectly rectangular silicon gratings (Fig. 1c).”

On page 3, there is a discussion of the effect of geometry on the Casimir effect, noting that most experiments are performed in the sphere-plate configuration. The authors should also note that experiments have been performed in the plate-plate (Bressi, Phys. Rev. Lett. 88, 041804, 2002) and sphere-sphere (Garrett, Phys. Rev. Lett. 120, 040401, 2018) configurations.

As suggested by Reviewer 1, we included the references on the plate-plate and sphere-sphere configurations, and added a sentence to the manuscript:

“Other configurations including plate-plate³⁵ and sphere-sphere³⁶ have also been measured experimentally.”

Figure 1: For the inset in (a), “ g ” is not clear. Also, for (c) and (d) and elsewhere in the manuscript, “displacement” is used to describe the separation between parts of the structure. The authors should clearly identify what is mentioned by “displacement” by assigning it a variable and putting it in the inset of (a) or by using a different variable for the x-axis, e.g. “ s ” which appears in the inset.

Following the suggestion from Reviewer 1, we changed the labeling of the lateral gap g in Fig. 1(a) inset and defined a variable d for the displacement of the movable grating in Fig. 1(b).

We edited the caption of Fig. 1 accordingly:

“The black dotted line denotes the initial location of the bottom edge of the blue movable grating. d is defined as the displacement of the movable grating from the initial position. “

For the PFA, it appears that the authors are assuming ideal metal plates rather than Si, which they use for the PAA. Is that the main difference? I don't think so, but the authors should explain a little more the difference between these two calculations because it appears that they are difference approximations about both geometry and optical properties. This will be helpful to make the conclusions of greater interested to the broader readership.

In comparing the PFA to the PAA, we always used the same structure, with the same geometry and the same optical properties (p-doped silicon). For example, Fig. 5(a) performs such a comparison for the device fabricated that has slightly rounded corners. Figure 5(b) does so for silicon gratings that are perfectly rectangular.

To avoid confusion, we modified the sentence in the main text:

“... compare it to the exact Casimir force calculated by SCUFF-EM in Figs. 5a and 5b for the silicon grating geometry of our device and the silicon gratings that are perfectly rectangular considered at the beginning of the paper, respectively.”

We also added the following sentence to the caption of Fig. 5 that compare results of PFA, PAA and SCUFF:

“The optical properties of silicon are used for all calculations.”

The main difference between PAA and PFA is how their basic elements are chosen. For PAA, the body of the structure is divided into small cubic blocks (we chose the volume to be 1 nm^3) and the pairwise van der Waals energy between the two groups of elements from the two gratings is summed. For PFA, the interacting surfaces are divided into small parallel plates that face each other. Then either the Casimir force or energy from the Lifshitz formula for each pair of parallel-plates is summed. The different ways to divide the interacting bodies into basic elements (PFA: small surfaces; PAA: small volume blocks) give entirely different results that only work well in a specific distance range.

Following the suggestion of Reviewer 1 to clarify the difference of the PAA from the PFA, we have added a section in Supplementary Notes 5 to discuss the details of the PAA algorithm to illustrate the difference from the widely used PFA.

As we discussed, all the results of PFA and PAA in the main text are calculated based on the real material properties, i.e. the property of Si. The only exception is when we discuss the dependence of the distance-independent force on the lateral separation g in the regime of interpenetration. We use perfect metals to obtain a g^{-3} dependence in the retarded limit. We added one sentence to clarify:

“So far, all the results of PFA presented are based on the real optical property of silicon used in the experiment. For simplicity and without loss of generality, we now consider rectangular gratings made of perfect metal separated by different values of g .”

Response to Reviewer 2

We thank Reviewer 2 for the recommendations to improve our paper. We are pleased that he/she considers our paper “suitable for publication in Nature Communications after authors clarify subtle points”

Reviewer 2 indicates that our paper needs the following clarifications.

Theory for evaluation of the Casimir energy between two aligned gratings in terms of Rayleigh reflection coefficients was developed in the paper [27] and in the paper with a lateral displacement of gratings: A.Lambrecht and V.N.Marachevsky, Int.J.Mod.Phys.A 24, 1789 (2009). This fact and these references should be mentioned since authors obviously use general expressions for the Casimir force in terms of Rayleigh coefficients in their calculations shown by purple squares on Fig.1 and Fig.5.

Following suggestions by the reviewer, we added the suggested reference in the manuscript.

What is the temperature T in the experiment (I could not find the value in the text) ? Why is it possible to use zero temperature calculations in comparison of the theory and experiment ? If authors use the last term in the righthandsight of the formula (4) at finite temperature (a Drude term), is there an agreement between the theory and experiment (did they check this) ? What is the difference between zero temperature and finite temperature results for the force ? It would be really good to comment on this in detail. Also it would be good to draw a plot with the ratio of zero and finite temperature theoretical results for the Casimir force at various separations.

The measurements were performed at a temperature of 4K (due to the need for the superconducting magnet for detecting vibration of the beam). Because the separation between the relevant parts of our structures is small ($\sim < 500$ nm), the thermal corrections to the Casimir force are negligible at 4K. We added the following text to clarify:

“To simplify the calculations, a temperature of 0 K is used instead of the actual temperature of 4K in the experiment. This approximation is justified because the separation between the relevant parts of the two bodies is smaller than 500 nm for all displacements. For example, at displacement $d = 1.6 \mu\text{m}$ where the top of the grating is about $0.3 \mu\text{m}$ from the main body of the beam, the calculated Casimir forces for 0 K and 4 K differ by $< 0.3\%$.”

The exact numerical results of Casimir force as a function of displacement at finite temperature (4 K) for our complex geometries take weeks to converge. Therefore, instead of generating a plot of the ratio of theoretical results for 4K and 0K as a function of d , we only manage to do the calculations for one value of d . For $d = 1.6 \mu\text{m}$, including finite temperature only changes the calculated force by $< 0.3\%$. The effect on the comparison of theory to measurement is negligible.

Authors should define the regions I, II, III, IV in nanometers and should emphasize how these regions are defined in general so that the readers could fully understand the boundaries between these regions. It is unclear to me where the regions II, III, IV start.

Following the suggestion of Reviewer 2, we added the following text to specify the regions:

“Specifically, region I ($d = 0$ to 430 nm) corresponds to the range of displacement before interpenetration. The measured force rapidly increases when interpenetration occurs in region II ($430 - 600$ nm). In region III ($d = 600$ to 1450 nm) the force is nearly independent of displacement. The force increases rapidly in region IV ($d > 1450$ nm) due to the interactions between the top of the gratings and the body of the beam.”

On line 189 it is written ~ 1000 , however, in all other places it is written ~ 500 . Probably there should be ~ 500 on line 189 as well.

The different factors correspond to two different geometries. For the ideal rectangular gratings with sharp corners, the deviation can reach 1000 times based on the numerical simulation from SCUFF and the semi-analytical scattering theory. For the device that is measured, the deviation is ~ 500 due to slight rounding of the corners in the fabrication process. In the abstract and conclusion, we use the experimentally demonstrated deviation of 500 times

We have therefore added the following text:

“In the inset of Fig. 4b the ratio between the measured force and the PFA shows a peak at a value of ≈ 500 at $d \approx 426$ nm which is weaker than the ≈ 1000 times deviation shown in the perfectly rectangular silicon gratings (Fig. 1c).”

It would be good to specify explicitly in the text the smallest distance between rectangular gratings for which authors were able to perform calculations within a scattering theory.

Theoretically, we could apply the scattering theory to our geometry as long as interpenetration does not occur, i.e. when displacement < 430 nm. In practice, when the structures approach each other, the time required for the calculations to converge increases drastically. With our limited computational resources, we apply the scattering theory only for displacement between 0 nm and 300 nm.

Following the suggestion from the reviewer, we added the following sentence to the manuscript:

“In principle, the scattering theory is applicable for displacements up to $d = 430$ nm when interpenetration occurs. However, calculations beyond $d = 300$ nm are beyond our

computation capability due to the computational power and time required for convergence.”

It would be good to discuss in more detail the difference between PAA and results of scattering theory. I expect it would be really interesting to add the plot where the ratio of PAA force to the exact force is explicitly shown at various separations.

Follow the suggestion of Reviewer 2, we add the ratio of the force calculated with PAA to scattering theory and numerical SCUFF results in Fig. 5c and the following paragraph to discuss the difference between PAA and results of scattering theory.

“The inset of Figure 5c plots the ratio of the forces calculated by the PAA to that by the scattering theory and SCUFF-EM for silicon gratings that are perfectly rectangular, in purple and red respectively. As expected, the purple and red results largely coincide with each other because both the scattering theory and SCUFF-EM works well in Region I before interpenetration (Fig. 1). At displacement of $0.3 \mu\text{m}$, the ratio is ~ 1.07 . The value decreases as the two gratings move farther apart.”

While discussing pistons in Discussion section I recommend the reference:
V.N.Marachevsky, Phys.Rev.D 75, 085019 (2007).

In this paper exact formulas for a piston with an arbitrary cross section were derived.

As suggested by Reviewer 2, we added the suggested reference.

In Supplementary information:

line 13 – should probably be “1 nm”,

We double checked that the scale bar measures $1 \mu\text{m}$. We re-wrote the sentence to avoid confusion:

“The scale bar in the main graph figure measures $1 \mu\text{m}$ (the width of the grating finger w is $\sim 900 \text{ nm}$). In the inset, the scale bar measures 100 nm .”

line 17 – should probably be “As discussed in the main text, ...”.

We thank Reviewer 2 for pointing out this typo. It is fixed in the revised paper.

Response to Reviewer 3

We thank Reviewer 3 for a thorough review and the thoughtful suggestions. We are pleased that he considers our paper “demonstrate that a significant challenge in the field of dispersion forces has been met” and “worth being published in Nature Commun” provided that “questions are answered.” We have followed his advice and made the corresponding changes.

The authors present an experiment that measures the van der Waals-Casimir force between two nanofabricated gratings of nearly rectangular shape that approach each other until they interpenetrate. The force gradient is measured by the frequency shift of a transverse oscillation of one of the gratings, and compared to numerically exact and approximate calculations. A good agreement with a fully numerical result (based on the code scuff-em) is found. Other approximations (like proximity force PFA or pairwise additivity PAA) fail in one or the other region of distances, for which the authors give heuristic explanations.

These results improve on the data shown in Ref. 14 by a team around H. B. Chan: the fabrication of the gratings is much better so that rectangular shapes are very closely achieved. The theory team has changed and is able to contribute calculations with different methods and approximations, although the full numerics is based on the same scuff-em code.

So why should this represent a significant advance to be published in Nature Communications?

First of all, to answer this question, we have adopted the following suggestion of Reviewer 3:

.....since it is quite obvious that the PFA is not applicable for this kind of system and since numerically exact calculations are available, it does not seem convincing to build the significance of the paper on the failure of an "out-dated approximation....."

But this is perhaps the main insight of this material: one can now fabricate structures where the PFA is utterly wrong (in some parameter region, as edges/corners slide against each other), and still one can demonstrate really good agreement between theory and experiment.

We agree with Reviewer 3 and have added the following text to address the above question:

(for previous experiments, including Ref. 14) “.....Even though the PFA cannot predict the Casimir force accurately in these experiments, it is computationally undemanding and is useful for a quick estimate of the order of magnitude of the force.”

(for the current experiment) “.....The experiment involves a number of improvements to the detection platform to enable the fabrication of structures in which, for a certain range of parameters, the PFA breaks down completely and fails to estimate the order of

magnitude of the Casimir force. There is good agreement between measurement and exact calculations using boundary element methods over the entire distance range, including the region where the PFA breaks down.”

In short, we have fabricated structures where the geometry dependence of the Casimir force is so strong that even the order of magnitude of the Casimir force cannot be estimated by PFA. It is necessary to run SCUFF-EM on a computer workstation for weeks (the red curve in Fig. 1) instead of obtaining it almost instantly by PFA.

We now address the other concerns of Reviewer 3.

A similar good agreement between experiment and theory was presented by Zou & al (Ref. 42) and Tang & al (Ref. 14).

While these two earlier experiments from our group have good agreement with theory, they actually did not demonstrate the geometry dependence (in the form of deviation from the PFA) like our current experiment. Ref. 42 measured the force between two beams that are nearly parallel. The exact force does not deviate much from PFA. In Ref. 14, the PFA only deviates from SCUFF-EM by ~40% (Fig. 1 in Ref. 14), and the measured force falls almost in the middle of these two values (SCUFF-EM and PFA). Ref. 14 acknowledged that “our measurement does not provide unambiguous evidence for the breakdown of the PFA.” In the present paper, on the other hand, significant improvements in the sample fabrication make it possible to create near-rectangular gratings that show the large deviations from the PFA. We added the following text to the introduction to clarify this point:

“A prior experiment measured the non-monotonic Casimir force when two T-shaped protrusions interpenetrate¹⁴. However, due to the limited resolution of optical lithography in the fabrication process, the protrusions are rounded at the corners. Moreover, there are non-uniformities among the different units, introducing uncertainties so that deviations from the PFA cannot be unambiguously identified. To our knowledge, the strong geometry dependence of the Casimir force in the regime of interpenetration for rectangular gratings remains unexplored.”

The fact that interpenetration leads to a constant Casimir force has already been mentioned and interpreted in earlier work, too, in particular in Refs. 14, 44.

To our knowledge, a constant and non-zero Casimir force was not discussed in Refs. 14 and 44. In Ref. 14 from our group, the force is not constant in any range of displacement. Even though the force in Fig. 1c is close to zero around 1.5 μm , it is actually changing rapidly with displacement, in a manner similar to two parallel plates. When the top of the protrusions on the

two sides are almost aligned, the force is not constant either, due to the significant rounding. The semi-circular shape of the top of the T-protrusions is given by the resolution of the lithographic process.

In Ref. 44, H.-C. Chiu and coauthors consider the lateral Casimir force between the sphere-plates configuration with sinusoidal patterns on both sides. The two surfaces never interpenetrate, and there was no measurement of a constant non-zero force.

To avoid confusion, we modified the sentence in the main text:

“...The measurement was performed when the two gratings were well-separated from each other without any interpenetration.”

The deviation with respect to PFA is exacerbated here (a relative factor of a few hundred), but it seems that this is due to a kind of simplistic way to apply the PFA, in particular when one deals with rectangles or rounded corners. In Ref. 14, the rounded corners of the T-shaped structures were taken into account with PFA (but perhaps it was rather the PAA that was applied there), giving a good qualitative agreement with scuff-em. So one may object that the "selling point of ~500" is an artefact of an oversimplified PFA.

[Clarification of PFA] The PFA is an approximation and is indeed simple to apply. As we discussed above, it can give rather accurate estimates on near-planar geometry and can estimate the order of magnitude of the Casimir force in all geometries in previous experiments so far [Ref. 14, 37-39, 45,46 etc], but not in the current experiment. To have a meaningful comparison to previous work, we use exactly the same standard PFA algorithm as in Ref. 14 (and also in other experiments). The interacting surfaces are divided into small pairs of parallel plates facing each other along the direction of displacement or perpendicular to it. The procedure can be applied regardless of whether the structures are rounded or near-rectangular. One may consider the PFA itself simplistic or oversimplified. But we made no attempts to further simplify it in our analysis. To our knowledge, deviations from the PFA remain the “figure of merit” of how strong the geometry dependence of Casimir forces is.

As reviewer 3 points out, in Ref. 14 the PFA is in good qualitative agreement with SCUFF-EM. Specifically, the deviation of SCUFF-EM from PFA is about 40% in Ref. 14 (even though measurement could not distinguish the two due to the non-uniformities in the different protrusion units). In this work on rectangular gratings, the deviations reach 500 times experimentally. The PFA is applied to the real geometry where the slightly rounded corners are fully taken into consideration in the same manner as Ref. 14. The large difference from SCUFF-EM originates from the geometry itself, and not the way that PFA is applied.

To avoid confusion, we added a sentence in the caption of Supplementary Fig. S1:

“All simulations on the real device, including SCUFF-EM, PFA, and PAA, are based on the digitalized boundary where the slightly rounded corners are taken into consideration.”

The authors should try to put their work into perspective in view of these remarks.

A few minor points are listed below. It may also be worth considering a shorter presentation of the experimental apparatus, since the same has been used in the Refs. mentioned above. The reader would rather learn more details about how the nearly perfect rectangular shapes have been manufactured: which processing steps led to success?

As suggested by Reviewer 3, we have shortened the description of the comb actuator as it has been discussed in Refs. 14 and 42.

Much effort was indeed spent on improving the fabrication process to yield the near-rectangular structures that are essential for the large deviation of the force from PFA. We agree with Reviewer 3 that more details should be provided.

We added a detailed process flow in Supplementary Note 3 to demonstrate how we achieve the near-ideal gratings. Briefly, we switch from optical lithography to electron-beam lithography for the fabrication of the rectangular grating parts, while keeping the optical lithography for the “big” structures such as springs and comb drives. Besides, we add one more mask layer (Poly-Si) on the top of the original oxide mask layer to transfer the pattern from e-beam lithography to the device layer with much better accuracy (thinner e-beam resist and better selectivity during plasma etching). Besides the two main points mentioned briefly above, we did a number of changes to improve the accuracy, such as changing the recipes of etching to be compatible with the new masks, adding a cooling step to the e-beam photoresist, optimizing the e-beam dose distribution based on the proximity effect correction from the Monte-Carlo simulation to achieve the better profiles, etc. The details can be found in Supplementary Note 4.

We also added the following sentences to the main text:

“A prior experiment measured the non-monotonic Casimir force when two T-shaped protrusions interpenetrate¹⁴. However, due to the limited resolution of optical lithography in the fabrication process, the protrusions are rounded at the corners. Moreover, there are non-uniformities among the different units, introducing uncertainties so that deviations from the PFA cannot be unambiguously identified. To our knowledge, the strong geometry dependence of the Casimir force in the regime of interpenetration for rectangular gratings remains unexplored.”

“In particular, a fabrication process involving electron beam lithography was developed (Supplementary Note 4) to yield highly precise rectangular structures with minimal rounding of the corners.”

If these questions are answered, then indeed, this makes an interesting paper that demonstrate that a significant challenge in the field of dispersion forces has been met. Worth being published in Nature Commun.

We have addressed all the questions from Reviewer 3. We believe that the modified manuscript warrants publication in Nature Communications.

p2, line 61-62: I think that Ref. 24 should not be over-advertised here. The heat transfer is actually ridiculously small, since it concerns just single modes in two membranes. And there are many other examples where the van der Waals force has mechanical consequences, as in the classical analysis of thin liquid Helium films.

Following the suggestion of Reviewer 3, we move Ref. 24 to the group of references on the operation of nanomechanical systems and removed the sentence on heat transfer to avoid over-advertising this paper.

line 67: "pair of plates" >"pairs of plates"

We have adopted this suggested change.

p4, line 99: the regime of interpenetration has been explored, indeed, in Ref. 14 (Tang & al.), just with a slightly different shape of the protrusions.

We have added a description of Ref. 14 and pointed out that interpenetration has been explored before.

“A prior experiment measured the non-monotonic Casimir force when two T-shaped protrusions interpenetrate¹⁴. However, due to the limited resolution of optical lithography in the fabrication process, the protrusions are rounded at the corners. Moreover, they are non-uniform among the different units, introducing uncertainties so that deviations from the PFA cannot be unambiguously identified. To our knowledge, the strong geometry dependence of the Casimir force in the regime of interpenetration for rectangular gratings remains unexplored.”

line 139: the "d" in the formula should be in italics

We have fixed this typo in the revised paper.

line 165: Casimir "path integral" -- I know that there are path integral formulations for the Casimir energy, but in the context of numerics like scuff-em, I don't think that this is an efficient starting point. Perhaps you mean "surface integral" which is taken, in this effective 2D geometry for the numerics, along a path, indeed. But the word "path integral" is fixed to Feynman's integral, please avoid the confusion.

To avoid confusion, we remove the phrase "path integral". The modified sentence now reads:

“SCUFF-EM calculates the force by evaluating the integral of Casimir energy using a classical boundary elements interaction matrix (see Methods for details).”

line 170: the sentence on the breakdown of PFA is a bit too much, that has been said a few lines before.

Follow reviewer's suggestion, we remove the phrase "the PFA breaks down" and change the sentence to:

“...while in region II, the PFA predicts an unphysical infinite force gradient.”

line 184: what is "N" in " $N = 100$ "? the number of Fourier modes? then just say so and drop the symbol.

We substituted the symbol "N" with "the number of Fourier modes"

Fig. 2(b): why repeat the sketches / micrographs with the interpenetrating fingers from Fig. 1? I know it's not a sketch, but it's still graphically a redundant information.

Following the suggestion of Reviewer 3, we have removed panels II and III of Fig. 2b. We note that Fig. 1 is only a sketch of a perfectly rectangular grating. It is not a real device. In Fig. 2(b), we need to keep panel I to show that the improved fabrication process can yield high quality rectangular gratings with slight rounding of the corners. We also choose to keep panel IV to show that the two gratings are well-aligned and the alignment is maintained even in the regime of interpenetration. These two panels are the result of much effort in creating rectangular gratings and minimizing lateral movements as the displacement increases.

caption Fig. 2, line 216: drop " $\omega_R =$ " or add the 2π that is needed here. The number of digits for the quality factor Q is ridiculous, I don't believe you can be so precise (same problem in the main text).

We have added 2π and used the appropriate number of digits for Q in both the caption of Fig. 2 and the main text.

p10, around line 220: I would like to know the number for the thickness of the facing gratings (along the z -direction).

The grating face each other has a thickness $\approx 2.58 \mu\text{m}$ which is measured from the cross-section of the device layer of the fabricated device. To avoid misleading, we change the sentence to:

“...the device that is fabricated using a combination of both electron beam and optical lithography on the $2.58 \mu\text{m}$ -thick device layer of a highly doped silicon-on-insulator wafer (See Methods).”

A simple basic question: it seems that the top beam (in red in Fig. 2) is "stiffer" than the lower grating (blue). Does that not mean that as the top beam is oscillating, it will "entrain" (via the Casimir force) the lower one. How can you be sure that the lower grating is kept in place and is not oscillating, too? Perhaps it is just a question of mass (the lower structure is much larger and more massive)?

We forgot to rotate the panels in Fig. 2b by 180 degrees before colorizing the figure. We thank Reviewer 3 for pointing out this mistake. In the corrected figure, the thin top beam is excited into resonance for detecting the force gradient. The wide lower beam is much stiffer and has a much higher resonance frequency. As both beams are strongly underdamped, the response of the lower beam at the resonant frequency of the upper beam is negligible.

line 261: correct spelling is "Torr".

line 308: replace V_o by V_0 (index zero)

We have corrected these two the typos.

p15, lines 319-21: "Thermal corrections are neglected as the zeroth Matsubara frequency term accounts for nearly all of the force" -- something is misunderstood here. The zero'th Matsubara term gives the *high-temperature limit* of the dispersion energy. You rather mean that the

integral over imaginary frequencies which deviates from the sum (i.e., the force at 4K) only by less than 0.3%. Or did I misunderstand something here?

We thank Reviewer 3 for pointing out this mistake. The revised sentence reads:

“To simplify the calculations, a temperature of 0 K is used instead of the actual temperature of 4K in the experiment. This approximation is justified because the separation between the relevant parts of the two bodies is always smaller than 500 nm. For example, at displacement $d = 1.6 \mu\text{m}$ where the top of the grating is about $0.3 \mu\text{m}$ from the main body of the beam, the calculated Casimir forces for 0 K and 4 K differ by $< 0.3\%$.”

p17, around line 363: the abrupt change in the PFA result seems like an artefact of having sharp corners (rectangular profile). If these corners are rounded (as shown by the TEM scan), then also the PFA can be modified to give a smooth result. This has been done in Ref. 14, for example. For sure, the discrepancy between experiment and "this version of the PFA" will be smaller, as also shown in Fig. 1 of Ref. 14.

Reviewer 3 stated correctly that rounding will reduce the deviation of PFA from the exact Casimir force. For example, the maximum deviation plotted in the inset of Fig. 1c for a rectangular grating with perfectly sharp corners (~ 1000 times) is larger than that for our device shown in Fig 4b (~ 500). In fact, this is the reason we spend the efforts to improve the fabrication to give near rectangular gratings.

However, we believe that describing our analysis as “this version of PFA” is inaccurate. We point out that the PFA used here is exactly the same as Ref. 14, as we explained in the paragraph [clarification of PFA] above. The large discrepancy between the measured force and the PFA originates from the rectangular grating geometry. It does not depend on the “version” of PFA used, because there is only one version.

The slightly rounded corners as shown in SEM images are already taken into account when we applied the PFA to the actual device. If the corners were perfectly sharp, the geometry becomes the perfect rectangular grating in which the force follows a step function with a vertical slope, as we plotted in Fig. 1c. With slight rounding of the corners, the slope is reduced as shown in Fig. 4b. However, as a function of displacement, the change in slope is still rather abrupt. We provide a more detailed explanation of the abruptness in the answer to the latter question by Reviewer 3 on Fig. 5.

line 365: the peak value 473 is too precise, I would only bet on something rounded like 500. You should be aware of your experimental errors for that ratio ...

Following the suggestion of Reviewer 3, we changed the peak value from 473 to ≈ 500 .

p17, line 374: the universal Casimir formula $\hbar c \dots / g^3$ is misleading here. As mentioned in my comments on the Supp Mat, one expects at the small distances here, that the energy per area rather follows the conventional Hamaker (nonretarded) formula A / g^2 (with material-specific Hamaker constant A). There is a problem with units in the Supp Mat calculation that leads to $1/g^3$.

In the analysis of the dependence of the distance-independent force on lateral distance g between grating fingers in the regime of interpenetration, we considered perfect metal instead of silicon to arrive at $1/g^3$ scaling in the retarded limit. We modified the following sentence to emphasize that perfect metal is used:

“So far, all the results of PFA presented are based on the real optical property of silicon used in the experiment. For simplicity and without loss of generality, we now consider rectangular gratings made of perfect metal separated by different values of g .”

We agree with Reviewer 3 that for real materials, smaller g will require the Hamaker formula. Therefore we have added the sentence:

“For gratings made of materials with finite conductivity, it is expected that for small g the scaling will change to $1/g^2$ in the non-retarded limit.”

We also edited the supplementary notes and fixed a mistake in Supplementary Note 2 as we will discuss later.

p18, around line 383: you mention in the details in Methods that the van der Waals potential used in the PAA assumes that there is no material between any two surface (even volume) elements. So it is intuitive to understand that this does not apply for the corners of the fingers when they interpenetrate, because they interact mainly across silicon (fingers in).

We agree with Reviewer 3 and have added the following description of the intuitive expectation:

“In the regime of interpenetration, it is more likely for material to be present between two interacting elements. Intuitively, it is expected that deviations from PAA are larger in region III compared to region I.”

line 390-91: the sentence "failure of PAA is from the non-pairwise additive nature" is redundant (french: pleónasme) and does not say anything.

Follow the suggestion from Reviewer 3, we have removed the sentence.

Fig. 5(a): I again would like to understand why the PFA gives a sharp onset, while in Fig. 1 of Ref. 14, the PFA starts off smoothly. Do your structures have so much sharper corners here?

In short, the answer is yes. The corners of the structures are indeed much sharper here compared to Ref. 14 because we create the structures with electron beam lithography instead of optical lithography. We added Supplementary Fig. S5 to compare the corners made by the two methods. Supplementary Notes 3 has also been added to describe the new fabrication process.

The main contribution of the PFA in region II and III comes from parallel plate elements that face each other in the x -direction. Before interpenetration, this contribution is exactly zero because there is no overlap between any of the parallel plate elements on the two sides. After interpenetration, the overlap area increases and so does the energy. For the gratings in this paper, the corners are only slightly rounded so that the transition from an exactly zero value to a non-zero value of this contribution is still abrupt. One can envision that with more rounding of the corners, such as to the extent in Ref. 14, the force predicted by PFA starts its rise smoothly after interpenetration.

line 430: replace "As the displacement is further increased so that the gratings interpenetrate each other" by "As the gratings interpenetrate each other" for those who only rapidly read the Conclusion.

line 437: typo "paths" > "paves (the way)"

line 456: I would expect "load the sample" rather than "load the probe"

We have adopted the above three suggestions from the reviewer.

line 469: I rapidly computed the plasma frequency for the given density of p-carriers and found a slightly different value, although I took the given effective mass:

We thank Reviewer 3 for pointing out the typo of the carrier concentration. We measured the sheet resistance and deduce the carrier concentration using Fig. 6.9 of [Pierret, R. F. Semiconductor Fundamentals, second edition, (Addison-Wesley, 1988).] The sheet resistance was stated correctly in our paper, while the carrier concentration should be $n \approx 6.0 \times 10^{18} \text{ cm}^{-3}$ instead. We have corrected it in the modified paper.

The correct carrier concentration gives the plasma frequency calculated from $\omega_p = e\sqrt{n/(\epsilon_0 m_{eff})} \approx 2.37 \times 10^{14} \text{ rad s}^{-1}$ and $\Gamma = \epsilon_0 \rho_s \omega_p^2 \approx 6.45 \times 10^{13} \text{ rad s}^{-1}$ shown in the method.

Methods, Eq.(8): well, this sounds like a horrible integral. Is there no way to find a reasonable approximation here? For the typical xi's in $\epsilon(\xi)$, unfortunately $c/\xi \sim 300 \text{ nm}$, comparable to typical distances, so you may not rely on simple power law approximations. But there must be reasonable Padé approximations for that, no?

The integration looks indeed complex and computationally expensive. However, in practice, we optimize our algorithm to speed up the calculation. With the optimization, the PAA calculation of the force as a function of displacement [Green lines in Fig. 5a,b] is completed within 20 minutes with our computation resources.

We added Supplementary Note 5 to describe details of our optimized algorithm where we reduce the $O(n^6)$ problem to $O(n^4+n^2)$. We have added the following sentence in the manuscript:

“Details of the algorithm for calculation of Eq. (8) for our geometry is presented in Supplementary Note 5.”

References: the citation style is not uniform (journal names in full or abbreviated, lower case or upper case). A number of references are incomplete:

We fixed the abbreviation error of references and the incomplete reference number for the Physical Review series.

p3, line 25: try to improve the ordering of the information. Suggestion: "between two ... corners, see inset in (a): two blocks initially separated by a gap of 80 nm (x-direction) and 500 nm apart in the y-direction. Main figures (a) and (b): calculated Casimir force and force gradient as ...

We have adopted this suggested change.

p3, line 32: if your bodies are infinite in the z-direction, you can only compute forces per unit length along that direction. So the units in Fig. 2 are wrong. Only the force per unit area, between perfect conductors, can scale like $1/g^3$, with the same exponent as the energy per area for two infinite plates (linear increase with the area as the bodies slide one against the other).

We thank Reviewer 3 for pointing out this mistake. We incorrectly stated that the objects considered in Fig. S2(a) are infinite in z direction. Instead, they are 2.58 μm thick.

The revised sentence reads:

“...each with cross-section of 3 μm by 3 μm square as shown in the inset of Supplementary Figure S2a and with thickness of 2.58 μm in the z -direction. The thickness is much larger than the size of the lateral gap (~ 80 nm).”

p4, line 42: I think that there is no "consistency with the simple PFA argument" of the main text.

(1) In the experiment, no exponent in the gap distance g can be found. (2) At the ~ 70 nm distance, it is likely that the Casimir energy in the experiment is already in the non-retarded (or Hamaker) regime, while for perfect reflectors, there is no non-retarded regime. Hence, the energy/area (and the force per unit length) scales like $1/g^2$. Since the length along the z -direction is constant, this scaling should also apply to the experimental force.

The confusion here again comes from the fact that we used perfect metal with finite thickness to get $1/g^3$ scaling of the displacement-independent force. Reviewer 3 correctly pointed out that the exponent is not measured in the experiment. We revised the text to emphasize that $1/g^3$ scaling applies to perfect metal only:

“This result is consistent with the simple argument on perfect metal using PFA in the main text that yields $F \propto 1/g^3$. If the gratings are made of materials with finite conductivity, it is expected that for small g the scaling will change to $1/g^2$ in the non-retarded limit. ”

REVIEWERS' COMMENTS

Reviewer #1 (Remarks to the Author):

They authors have sufficiently address my concerns, and in my opinion the paper can be published in Nat Comm.

Reviewer #2 (Remarks to the Author):

I have read the revised version of the manuscript and comments of the authors. I am satisfied with authors' reply to comments and changes/additions made in the manuscript.

I have noticed that in Refs. [22], [40] "Casimir" is written as "casimir". Capital letter "C" should be used in "Casimir" in titles of these references.

I already wrote about achievements of the manuscript in detail in my previous report. I recommend this manuscript to publication in Nature Communications after correction of the issue above.

Reviewer #3 (Remarks to the Author):

Strong geometry dependence of the Casimir force between interpenetrated rectangular gratings by M. Wang, L. Tang, C. Y. Ng, R. Messina, B. Guizal, J. A. Crosse, M. Antezza, C. T. Chan, and H. B. Chan
submitted to Nature Commun
ms# 273817-1

I am happy with the revisions made by the authors. They clarified certain points related to the substantial advances made here compared to earlier work. As stressed in my earlier report and in those of the other referees, this experiment illustrates the high degree of quantitative understanding that is now available for dispersion forces in sub-micron scale, complex geometries. There is no doubt that this topic attracts a wide inter-disciplinary audience. I recommend publication.

Minor typos

Main text

When calling references in-line ("as shown in Ref. xx"), avoid typesetting xx as superscript (use LaTeX "\onlinecite{key-to-xx-ref}").

p1, line 39: suppress "nanoscale" (kind of redundant)

p5, line 120: insert "[to estimate] even [the order of ...]"

p6, line 143, caption Fig. 1: replace "[from] this [initial position]"

p7, line 167: correct is "become"

p9, lines 198, 199: suppress the abbreviations FMM and ASR, as they are never used in the main text (sorry, Brahim Guibal).

p17, lines 359, 360: typeset symbol d in italics

p18, line 397: correct is "[real optical] properties"

p20, line 440: correct is "work [well]"

p25, line 538: correct is "[Details ...] are [presented]"

p29, line 628, Ref. 40: Capitalize "Casimir" (in bib database: {C}asimir)

Supplementary

p8, line 132: write "U_c" with "c" or "C" as sub-script (same p9, line 144)

p9, line 142: typeset all n's in italics

Carsten Henkel

Response to Reviewer 1

They authors have sufficiently address my concerns, and in my opinion the paper can be published in Nat Comm.

We are pleased that Reviewer 1 recommends publication of our paper.

Response to Reviewer 2

I have read the revised version of the manuscript and comments of the authors. I am satisfied with authors' reply to comments and changes/additions made in the manuscript.

I have noticed that in Refs. [22], [40] "Casimir" is written as "casimir". Capital letter "C" should be used in "Casimir" in titles of these references.

I already wrote about achievements of the manuscript in detail in my previous report. I recommend this manuscript to publication in Nature Communications after correction of the issue above. They authors have sufficiently address my concerns, and in my opinion the paper can be published in Nat Comm.

We are pleased that Reviewer 2 recommends publication of our paper.

Following the advice of Reviewer 2, we have put capital C for “Casimir” in Refs. [22] and [40].

Response to Reviewer 3

Strong geometry dependence of the Casimir force between interpenetrated rectangular gratings

by M. Wang, L. Tang, C. Y. Ng, R. Messina, B. Guizal, J. A. Crosse, M. Antezza, C. T. Chan, and H. B. Chan

submitted to Nature Commun

ms# 273817-1

I am happy with the revisions made by the authors. They clarified certain points related to the substantial advances made here compared to earlier work. As stressed in my earlier report and in those of the other referees, this experiment illustrates the high degree of quantitative understanding that is now available for dispersion forces in sub-micron scale, complex geometries. There is no doubt that this topic attracts a wide inter-disciplinary audience. I recommend publication.

Minor typos

Main text

When calling references in-line ("as shown in Ref. xx"), avoid typesetting xx as superscript (use LaTeX "\onlinecite{key-to-xx-ref}").

p1, line 39: suppress "nanoscale" (kind of redundant)

p5, line 120: insert "[to estimate] even [the order of ...]"

p6, line 143, caption Fig. 1: replace "[from] this [initial position]"

p7, line 167: correct is "become"

p9, lines 198, 199: suppress the abbreviations FMM and ASR, as they are never used in the main text (sorry, Brahim Guibal).

p17, lines 359, 360: typeset symbol d in italics

p18, line 397: correct is "[real optical] properties"

p20, line 440: correct is "work [well]"

p25, line 538: correct is "[Details ...] are [presented]"

p29, line 628, Ref. 40: Capitalize "Casimir" (in bib database: {C}asimir)

Supplementary

p8, line 132: write "U_c" with "c" or "C" as sub-script (same p9, line 144)

p9, line 142: typeset all n's in italics

Carsten Henkel

We are pleased that Reviewer 3 recommends publication of our paper. We have corrected all the typos listed by Reviewer 3 in both the main text and the supplementary information.